# Public Health Directives in a Pandemic: Paradoxical Messages for Domestic Abuse Victims in Four Countries

**DOI:** 10.3390/ijerph192114148

**Published:** 2022-10-29

**Authors:** Soma Gregory, Stephanie Holt, Christine Barter, Nicola Christofides, Ogopoleng Maremela, Nobulembu Mwanda Motjuwadi, Cathy Humphreys, Ruth Elliffe, Nicky Stanley

**Affiliations:** 1School of Social Work and Social Policy, Trinity College, University of Dublin, D02 PN40 Dublin, Ireland; 2School of Social Work, Care and Community, University of Central Lancashire, Preston PR1 2HE, UK; 3School of Public Health, Faculty of Health Sciences, University of the Witwatersrand, Johannesburg 2000, South Africa; 4COPESSA (Community-Based Prevention and Empowerment Strategies), Johannesburg 1818, South Africa; 5Department of Social Work, University of Melbourne, Victoria 3010, Australia

**Keywords:** COVID-19, domestic abuse, public messaging, national messaging, awareness raising

## Abstract

When the COVID-19 pandemic manifested urgent concerns were raised around the globe about the increased risk that public health restrictions could pose for victims of domestic abuse. Governments, NGOs and community services swiftly responded to convey the message that services for victims were operational and restrictions did not apply to those fleeing harm. This paper reports on the various approaches used to communicate this public health messaging during COVID-19, further highlighting strengths and learning which could inform future crises messaging. It utilises data gathered through a rapid review and mapping of policy and practice initiatives across 4 high-middle income countries: UK, Australia, South Africa and Ireland. Four themes were identified: (1) Top-down: National media messaging; (2) Top-down: Political leadership; (3) Traditional media vs. social media and (4) Bottom-up messaging: Localised, community-based messaging. It was found that a strong, clear top-down stance on domestic abuse was perceived as beneficial during COVID-19. However, a stronger focus on evaluation, reach and impact, particularly for minority groups may be required. Newer forms of media were shown to have potential in conveying messaging to minority groups. Community and grassroots organizations demonstrated their experiential knowledge in reaching target audiences. Harnessing this expertise for future crises messaging may be valuable.

## 1. Introduction

Intimate violence, including domestic abuse (DA), is a pressing public health concern. The term DA is term widely used in English speaking European countries such as Ireland and the UK. This term, rather than emphasizing physical abuse, encapsulates the range of abusive behaviours experienced by victims and survivors (e.g., verbal/emotional, financial/economic, psychological, sexual, cultural/identity and coercive control). Estimates published by the World Health Organisation (WHO) [1] indicate that globally 1 in 3 women have been subjected to either physical and/or sexual violence in their lifetime, whether perpetrated by an intimate partner or non-partner. DA has a significant impact on the health and wellbeing of adults and children across the world, accounting each year for over half (58%) of all female homicides [2] and over 16 million cases of non-fatal violence-related injury severe enough to require medical attention [3]. Women who have been physically or sexually abused report higher rates of health problems, are 16% more likely to have low birth weight children, are more than twice as likely to have an abortion, are almost twice as likely to experience depression and, in some regions, are one and a half times more likely to acquire HIV than those who have not experienced abuse [3]. This type of violence leads to high social and economic costs for families and societies [1]. For example, the estimated cost of DA in England and Wales for year ending March 2017 was approximately £66 billion per year [4]. The UN sustainable development goals targets specifically refer to intimate violence prevention as an essential aspect of development [5] However, since the pandemic and accompanying restrictions, these issues have become an even more crucial public health concern requiring a crisis response to protect victims and survivors of DA, in addition to other identified vulnerable populations [1]. While women in heterosexual relationships comprise the vast majority of DA victims; it must also be acknowledged that other groups, including men, those in same sex relationships and children and young people exposed to violence or abuse, can also be victims. On this basis, the term DA will be used to include the range of victims experiencing interpersonal violence.

Awareness raising is a critical component of violence prevention strategies and is promoted by The European Institute for Gender Equality (EIGE) as being both an efficient and effective method of imparting essential information to the public [6]. It has been widely used as a tool for combatting all forms of violence against women and Article 13 of the Council of Europe Convention on Preventing and Combatting Violence against Women and Domestic Violence (the Istanbul Convention) requires member states to undertake extensive awareness raising initiatives as part of a comprehensive suite of measures targeted at the prevention of violence against women. As such, DA campaigns are used to prevent victimisation, avert perpetration of DA, supply information about DA resources and supports, and communicate the core principle that DA is an unacceptable violation of human rights [6].

When the COVID-19 pandemic manifested in early 2020, urgent concerns were raised around the globe about increased levels of DA and the risks that public health strategies and restrictions, instigated to curb the transmission of COVID-19, could pose for the safety of victims [7,8,9]. Paradoxically, the ‘stay-at-home’ public health directives intended to shield, were in fact potentially harmful to those living with DA. Worldwide, governments initiated responses to specifically address these concerns. Traditionally, government action on public health issues can include large scale awareness raising, public messaging campaigns [10] and speeches from political leaders [11], with government messaging considered critical to informing attitudes and behaviours, and minimising harm [12]. The official core message conveyed about DA during COVID-19 was that help was still available, services were still operating, and more importantly that any restrictions on movement did not apply to those seeking support or refuge. These campaigns promoted awareness of DA supports and resources, aiming to facilitate the help-seeking process [13,14]. Running parallel with awareness raising campaigns and mass media messaging, many nations also conveyed messages via political leadership, which has been described as a ‘critical pillar’ [11] of influence in matters of public health, with potential to influence public attitudes and behaviours. There is however limited evidence on the impact and influence of political leadership in respect of attitudes and behaviour concerning interpersonal violence in crisis situations such as the COVID-19 pandemic.

In this article we report on the approaches used to deliver national public messaging during COVID-19 across four high-middle income countries aiming to identify strengths and learning which could potentially inform future national public messaging initiatives.

### Background

Public health campaigns to promote healthy behaviour have an established track record [15]. Mass media campaigns that utilise combinations of media formats can be a relatively inexpensive way to reach large audiences [15] whilst also simultaneously targeting segments of that audience [15,16]. While national television, radio and print media have been the traditional vehicles for the dissemination of these messages, the use of social media and cell phones is a more recent popular and effective communication strategy, particularly at grassroots and local community levels [17,18]. It has been suggested that social media offers an important pathway for sharing information, as posts can be widely dispersed, and have also been found to be effective in accessing hard to reach groups [16]. A heightened awareness of this has motivated DA agencies to tailor and target messaging to maximize impact [16], connecting victims and their support networks [19]. Social media has been used in the dissemination of health information during periods of public health crisis and for sexual health promotion [20]. As a ‘real-time network’, social media provides both public and private messaging capacity [16], with Twitter one of the most widely used platforms for garnering attitudes and beliefs on public health concerns while also communicating on social issues. To illustrate, 90,000 Twitter users responded to the release of the video in 2014 of National Football League (NLF) player Ray Rice punching his fiancée, using the hashtag #WhyIstayed [21]. Similarly in 2017, Twitter was a major forum for conversations about the #MeToo movement. Instagram, a social networking platform largely based on video and photograph sharing, also uses caption, description and hashtag to highlight or reinforce the messages conveyed [18].

As mobile phones have the capacity for mobile-to-mobile sharing of video content, they can provide fast, accessible and low-cost messaging as part of awareness raising campaigns [18]. A further attraction of this medium relates to the limited use of conventional media channels in reaching all segments of the population [15]. Remote geographical areas may lack television or radio stations, or those available might operate with weak to no signal coverage. However, in these areas, mobile phone use is often prevalent, and using this medium for dissemination of health promotion campaigns has clear potential as a communication strategy, especially among marginalised communities that are difficult to reach using more conventional channels [22].

A number of studies focus on the importance of awareness campaigns and public messaging processes being accessible to minority groups [13,23]. Accessibility of messaging relating to DA is particularly important, as excluded or minority groups may be at increased risk of experiencing DA [24], and also face multiple barriers accessing services, whether because of lack of English language proficiency [13,24,25], services that are unresponsive to their needs [24,25,26], or fear of police [13,25]. Moreover, the COVID-19 pandemic has reinforced existing inequalities [24], and there have been warnings that sexual [27], gender [23], racial and ethnic minorities [25,28] have been disproportionately impacted by the pandemic.

Participants in a UK study reported the lack of DA prevention materials for LGBT young people, with the authors concluding that the specific needs of this group may be particularly urgent as emotional abuse of partners may be compounded by the threat of unwanted “outing” [29]. Similarly, a national transgender survey highlighted disproportionate rates of DA in the Latinx community in the US, with rates particularly high for transgender men, the unemployed and the disabled, with structural and interpersonal challenges experienced by these marginalised groups being identified as compounding factors [30]. Another US study on DA help-seeking in the Lantinx community during COVID-19 highlighted the need for awareness raising campaigns to address diversity in their messaging through better use of ‘gendered language, images, and stories’ [13]. The authors also identified the value of conveying messaging in accessible language and signposting resources provided by and for minority communities who might fear contacting the authorities [13].

The key hypothesis underpinning mass media campaigns is that delivering a message to whole populations will result in changed behaviour consequent on new information or changed attitudes [31]. However, it is known that social and health related behaviour change can be difficult or complex to achieve [32]. Additionally, there is a paucity of information elucidating how exactly and to what extent media campaigns in particular effect change in violent and abusive behaviour [29,33,34]. There are methodological challenges and ambiguity around defining and measuring the effectiveness of such campaigns [32] and evaluation requires an understanding of the processes of change, particularly for groups and individuals where abusive behaviour is ‘embedded and habitual’ [29]. It has been argued that ‘evaluation must be embedded into campaigns at the onset’ if the aim of the campaign is to change behaviour [29].

Lastly, given that generic national campaigns deliver general messages collectively to many diverse groups; the design of these campaigns needs to ensure that the messages are specifically designed to ‘reach-and to be heard by—the people that need to hear them’ (p.20) [35]

## 2. Materials and Methods

This paper derives from a wider study exploring DA policy and practice for survivors, children and perpetrators during the COVID-19 pandemic across four countries—the UK, Australia, Ireland and South Africa. These were selected as high-middle income countries with established DA services which were at different stages of development. The study aimed to capture and assess policy and practice initiatives in the four countries in responding to domestic abuse under COVID-19 with a commitment to examine both ‘top-down’ and ‘bottom-up’ initiatives.

To this end, a mapping study of policies and practices that were implemented during this time was undertaken in each country. This involved a rapid review of evidence gathered via key stakeholders that held membership of relevant professional, policy, research and practice networks. Following this ‘Call for Evidence’, expert interviews were undertaken in each country to supplement the data gathered. Data was extracted into a bespoke spreadsheet which was used across all four countries. Additionally, a simple data appraisal tool, developed from questions used in an European Institute for Gender Equality 2021 study [36], was amalgamated into the spreadsheet. Analysis used a common framework based [37] on key research questions across all four countries with some local variations. Findings from each country’s mapping study informed the selection of an in-depth case study into a specific DA related policy or practice of note from this period. For the case studies, evidence from the mapping studies was utilised, and was supplemented by additional focused data collection, including further expert consultations and interviews where required.

The focus for the Irish case study [38] was Ireland’s national awareness raising campaign, devised and delivered during COVID-19 to inform both female and male victims that help was still available and accessible. We draw on the findings of Ireland’s case study and consider them alongside the different approaches taken to public health messaging in the UK, South Africa and Australia. For the purpose of this article, each participating country provided their data on national messaging, as it was approached in their country. After key areas were identified across the four nations, additional supplementary data was extracted from the evidence collected for the rapid review mapping studies conducted within each country, as set out above. Following extraction, the data was thematically analysed.

## 3. Results

Four main themes were identified from the analysis: (Section 3.1) Top-down: National media messaging; (Section 3.2) Top-down: Political leadership; (Section 3.3) Traditional media vs. social media; and (Section 3.4) Bottom-up messaging: Localised, community-based messaging.

### 3.1. Top-Down: National Media Messaging

Government directives that instructed people to remain in their homes in an effort to slow the spread of the virus were introduced in all four countries early in the first phase of the pandemic. These ‘stay at home’ directives were later understood to have been detrimental to those experiencing DA:


*“So, I think we were very slow to recognise the literal way in which people were interpreting that [message]”*
(UK Interview 7—Scotland)

In South Africa, the directive to ‘stay at home’ was judged to have increased women and children’s exposure to DA; the combination of overcrowding in households as migrant workers returned, the consequent loss of income from lockdown measures and food shortages, together with a ban on the sale of alcohol and cigarettes, aggravated already high stress levels in households due to COVID-19 [39]. In Ireland, participating stakeholders from the DA sector revealed that initially there were misunderstandings and confusion over whether the 2 km/5 km travel restriction imposed by the government applied to those fleeing DA [40].

In all four countries, DA organisations highlighted the increased risks that adherence to ‘stay at home’ directives could pose for DA victims, and governments stepped in to provide clarity around messaging using a range of media channels to convey that help was still at hand; essential services and supports were open and operational, and furthermore that victims were exempt from any travel restrictions or curfews in place. It was reported that these campaigns, and the heightened media coverage they leveraged, may have had an impact on increasing awareness of DA not only for the general public, but also in private and government sectors [41].

We found many commonalities between the approaches taken to national media campaigns across the four countries. Traditional television, radio and print media were supplemented by online resources and social media posts to enhance visibility of the campaigns. The UK campaign launched under the hashtag #YouAreNotAlone, ‘At home shouldn’t be at risk’, in April 2020 [42]. Ireland launched the ‘Still Here’ campaign on 15 April 2020 [43]. The Australian federal government developed the information campaign ‘Help is Here’ which commenced on 3 May 2020 [44], with the UK and Australia additionally supplementing these messaging endeavours with materials in supermarkets and shopping centres. The South African government did not develop a national media campaign, instead this task was undertaken as part of ‘The Solidarity Fund’, a philanthropic/corporate joint venture which was designed as a rapid response to fight the health, humanitarian and social consequences arising from the COVID-19 pandemic [45]. Funding was made available to specifically address gender-based violence during the pandemic. As such, the Solidarity Fund provided humanitarian funding to both government and nongovernmental organisations working with DA [46]. The Solidarity Fund campaign commenced on 15 November 2020, and comprised of radio, print, digital and social media. It utilised national African language stations to reach as many people as possible [46].

A key strength of these national campaigns was that they were large budget operations transmitted through mainstream media channels in the UK, Australia & Ireland. Stakeholders consulted generally perceived these campaigns as providing a crucial spotlight on the issue of DA during the pandemic. It was considered that the combination of national media campaigns, together with increased press attention, created heightened public awareness around the issue of DA and fostered a greater empathy for victims. In some countries, such as Ireland, stakeholders perceived that increased empathy and heightened awareness may have fuelled higher levels of private donations to support DA services during the pandemic. Reflecting on this, one stakeholder noted that increased attention on DA:


*“Sort of gave it a solo spotlight over that sort of 8, 9, 10 months where people said ‘Okay, DV is a really important thing’ ”*
(Stakeholder 5—Ireland)

‘Help is Here’ [47] was the Australian government’s DA awareness campaign developed by Department of Social Services (DSS) during COVID-19, which sought to inform victims that help was still available during pandemic restrictions. This campaign was the only campaign identified by this study which also targeted perpetrators, urging them to seek support to change their behaviour. As with all government awareness campaigns in Australia, the ‘Help is Here’ campaign was obliged to comply with the government’s campaign advertising framework principles, this means that campaigns of this nature are subject to specific review, certification and publication requirements. Government campaigns are formally appraised by the Australian National Audit Office (ANAO) for compliance with the guidelines. Under this process, government funded advertising campaigns must also be evaluated for effectiveness.

The ‘Help is Here’ campaign was devised using a trauma-informed approach. As such, the imagery portrayed was of genderless characters and did not depict any scenes which could be ‘triggering’ or that might have an adverse impact on vulnerable audiences [48]. Wave 1 ran from 3 May to 3 October 2020. Wave 2 ran from 1 November 2020 to 30 January 2021. Advertising activities comprised of television, radio, newspapers, magazines, out-of-home digital banners and digital displays across both waves. During Wave 1 of the campaign there was also brand partnership integration and public relations activities. The campaign evaluation report noted that during Wave 1 of the campaign both awareness and use of helpline numbers increased from benchmark figures [48]. Helpline figures were measured for the 1800RESPECT phoneline, which is the national domestic, family and sexual violence counselling service, and for MensLine Australia which offers men specific counselling services including family violence and perpetrator supports. The campaign also provided regional DA support information.

In order to comply with campaign advertising framework guidelines, targeted campaign materials were produced. The campaign was translated from English into 14 different languages; Arabic, Cantonese, Farsi, Greek, Hindi, Italian, Korean, Mandarin, Punjabi, Spanish, Tagalog, Tamil, Thai and Vietnamese in order to reach Culturally and Linguistically Diverse communities (CALD). Aboriginal and Torres Strait Islander Australians were targeted by adapting the mainstream advertising and utilising Indigenous-specific communications providers [48]. Those living in remote or rural areas were targeted using regional media channels. All campaign advertising was closed captioned and Auslan (Australian Sign Language) versions were also developed.

The ANAO reported that they were advised by the DSS in October 2021 that “monthly contacts to 1800RESPECT increased during the Help is Here campaign starting at 23,555 contacts in April 2020, peaking at 31,734 contacts in August 2020 and tapering back to 21,629 contacts in February 2021. The demand for MensLine services was variable, but overall remained strong, during the ‘Help Is Here’ campaign” [48]. Furthermore, the evaluation report noted that the launch of the campaign saw significant increases for key actions such as “visiting the website, thinking about accessing support and making enquiries about available support” (p.94) [48]. In addition to uptake in services, the DSS monitored week-to-week changes in attitudes towards DA, as well as knowledge of DA support services during Wave 1, and these were monitored monthly during Wave 2. A proposed Wave 3 for the campaign did not go ahead as helplines did not report any further increases towards the end of the Wave 2. Australia is the only country in this study to use detailed guidelines to regulate government advertising campaigns. This approach provides an evidence base for campaign effectiveness and success in reaching their target audience as well as a rationale for repeating such campaigns.

Elsewhere, while perceptions of national campaigns addressing DA were reported to be extremely positive by participating stakeholders, there was limited robust evidence on impact and reach for this type of messaging. In Ireland, the case study work [38] described above provides some detail on the evaluation of the ‘Still Here’ campaign. Two separate formal evaluations reported that recall was high for this campaign: a Department of Justice commissioned nationally representative survey of 1000 adults found that 86% could recall the campaign [49]. Similar results were reported amongst participants in the second survey which was commissioned by a women’s refuge [50]. Moreover, both evaluations found that the national campaign was perceived to be ‘effective’ and ‘realistic’ in communicating its message by participants in both surveys [49,50]. More tangible metrics, such as uptake of services arising from the campaigns were not available. However, it was concluded that a primary strength of this campaign was its strong, clear association with the official policing response to DA during COVID-19: ‘Operation Faoiseamh’, which translates to ‘Operation Relief’ in English. The ‘Still Here’ campaign website contained links to DA specific supports and services, including to An Garda Síochána (the Irish police force). In addition to law enforcement measures, Operation Faoiseamh featured its own media component which communicated a ‘zero tolerance’ stance on DA to both perpetrators and the wider community, and reassured victims that the police were available to provide assistance at any time during the pandemic. Operation Faoiseamh was perceived to have been pivotal in addressing DA during the pandemic in Ireland, as illustrated by the following quote from an Irish stakeholder:


*“[on Operation Faoiseamh] we cannot underestimate how powerful that was. It was powerful because it was named. It was also powerful because Garda [the police] were enabled and empowered to do the work that they really actually want to do. And perpetrators were on notice that they would be identified and pursued. And that meant a huge amount”*
(Stakeholder 7—Ireland)

As far as could be ascertained, no evaluations were undertaken in the UK of the #YouAreNotAlone campaign.

The issue of reach was raised in the UK and Ireland, with study participants reflecting that the campaigns would have increased reach had the messaging been available in a variety of languages (as with the Australian campaign) or if campaigns had also specifically targeted more marginalised groups. Additionally, further issues relating to reach were highlighted in the UK where it was reported that leaflets and other materials on DA were only made available in some vaccination centres and COVID-19 testing locations. It was indicated during interviews that the rationale behind this decision involved concerns that distributing DA messaging in these contexts may have ‘diluted’ safety messaging around COVID-19, or possibly deterred individuals from getting their vaccinations, which was viewed as the priority. However, it was concluded that the decision not to promote supports and services for victims and survivors in these locations was a missed opportunity. Indeed, the UK findings suggested that prevention awareness raising should perhaps fall under the remit of public health, who could play a more active role in the development, execution and evaluation of campaigns to maximise their potential. Nevertheless, participants recognised the impact of high-profile campaigns in their countries:


*“So, one of the things for me is those public awareness campaigns run by them [the Welsh government] what they were really keen to do, was to just push that message out there of, ‘you are not alone and home shouldn’t be a place of fear’. So, they were much broader, they weren’t sort of focusing on one space. They were just sending that message out there, out there, out there.”*
(UK Interview 4—Wales)

### 3.2. Top-Down: Political Leadership

Communication from political leadership was another key component in national messaging and this was utilised by all governments covered by this the study to varying degrees. Where this approach was used, it was generally seen to demonstrate leaders’ understanding of the issues experienced by victims of DA. It also was perceived to have represented a commitment to addressing the crisis, providing reassurance that help was available to victims and survivors. In making DA victims an exception to the general COVID-19 restrictions, governments clearly identified them as a group entitled to special consideration and protection. In South Africa, DA was already high on the political agenda; the COVID-19 lockdown struck at a time when a process to develop and implement a national strategic plan to combat gender-based violence and prevent femicide had been in place for the preceding two years. In place of a national messaging campaign, President Cyril Ramaphosa made emergency addresses to the nation throughout the pandemic. During the period from 24 March to 31 December 2020, President Ramaphosa made 19 speeches addressing the lockdowns and restrictions to curb the transmission of COVID-19. These speeches were broadcast on multiple channels including all major television channels, radio stations, news media (both online and print), social media and YouTube. Twelve of the 19 (63%) COVID-19 focused speeches mentioned violence against women and children perpetrated by men. As the extract below from one of the speeches illustrates, a clear link was made between the pandemic and DA, with the home being identified as a space of danger and potential violence:


*“The coronavirus pandemic heightens the risk of gender-based violence as women may be experiencing emotional and physical abuse behind the walls of their homes.”*
(President Ramaphosa, 30 April 2021)

There were diverse reactions among civil society participants interviewed regarding the impact and value of the President’s focus on DA. Some felt that it emphasized the seriousness of the issue and that it raised awareness among certain sectors within communities, especially men. Others believed that the speeches happened too infrequently to have meaningful impact and that while there was value in expressing commitment, there needed to be more intense awareness-raising. There was some scepticism expressed that the speeches were primarily about politics rather than representing a deep commitment to DA:


*“I respect our president, but sometimes, the politicians talk the talk, but they don’t walk the walk. You know, they do a lot of promises, which we haven’t seen happen yet. Our budget gets cut most of the time every year. It’s good to tell a country of how many women died and what just happened that happened, but we haven’t seen it at ground zero”*
(Interview S5—South Africa)

In England, study participants noted that only very rarely was messaging aimed at victims during the daily televised UK Government briefing. Many interviewees commented that this was a missed opportunity to emphasize that the directive did not apply if home was not a safe place and reinforced that help was still available. In contrast, interviewees from Scotland and Wales generally felt their First Ministers had taken a lead role in DA messaging, referring to DA regularly at daily press conferences and through social media. This suggests that stakeholders valued a strong public government stance on DA with top-down messages interpreted as an important display of solidarity:


*“And I’ve noticed, even sort of the last six months, ministers are much more keyed up on these issues and they’re much more interested”.*
(UK Interview 10—Scotland)

### 3.3. Traditional Media vs. Social Media

In the UK, Ireland and Australia, increased traditional media attention (TV, radio, press) on DA during the pandemic was highlighted as pivotal to increased public awareness on the issue. Some commentators suggested this increased attention occurred because other crimes diminished during lockdown, and therefore DA had become the crime of focus resulting in greater media attention [51]. Stakeholders interviewed commented that, as the public’s freedom was curtailed and restricted, it became easier for them to imagine what life might be like in a coercively controlling relationship and to be trapped at home, especially if home was not a safe space. Interviewees considered that increased journalistic coverage had been helpful in providing wider messages to the public and DA survivors, sparking a national debate, or opening up a national conversation:


*“I think the positive side of Covid is that domestic abuse, and particularly coercive control… has actually come to the fore hugely, partly because… we’re all imprisoned really and how we react has become much more to the forefront of media attention.”*
(UK Interview 21—England)

Although some stakeholders questioned whether this media attention reached its intended audience, or whether it was ‘heard’ by those who might need to hear it:


*“There were lots of media articles about DV. The media is interested. But does that filter down to women in the suburbs living with DV? I don’t know. There was a lot of talk about women going into lockdown, and survival.”*
(Australian Interview 7)

In Australia, the focus on DA was already reported to have waned within the period of the study; with the term ‘shadow pandemic’ that widely referred to DA early in the pandemic, being reappropriated to describe high levels of mental health need, raising questions about the sustainability of awareness raising endeavours in traditional media. This sentiment was echoed by participants in the UK study where DA organisations and DA commentors expressed concerns about whether the traditional media’s interest in DA would endure in the longer term.


*“My concern is what comes after and …as the press slides away, whether some of these …improvements actually lead to real change. We need change in convictions, we need change in support … for perpetrators. We need more resources, all of that. And whether that then still has the interest of the press, I don’t know.”*
(UK Interview 22—England)

However, it was also recognized that traditional media may not have reached all groups. It was pointed out by several stakeholders in the Irish and UK studies that these media stories were only delivered in English and so might not reach victims who lacked language proficiency. A representative from a DA service working with minority ethnic women in Ireland emphasized that language was a ‘big barrier’, further highlighting that these groups tended not to watch, listen or read traditional mainstream English language media, preferring to use social media or the internet to access media broadcast in first languages. Survey research conducted during this time by Muslim Women Australia (MWA) reported that the primary source of information for communities from CALD backgrounds was social media (38%) [50]. A number of stakeholders across the UK, Australia and Ireland highlighted the strength of social media in targeting minority groups, when a variety of languages were required, or where content needed to be targeted in a culturally sensitive or appropriate way. Examples included the Traveller community in Ireland, the Black Asian Minority Ethnic (BAME) community in the UK and the CALD community in Australia. It was highlighted by stakeholders that services and organisations working directly with these groups were best positioned to advise how to target and reach specific groups.

### 3.4. Bottom-Up Messaging: Localised, Community Based Messaging

A key feature across all four countries was the swift and decisive action taken by the DA sector, together with grassroots or community organisations in responding to the restrictions imposed by the pandemic. These organisations demonstrated their expertise and experiential knowledge of how to reach their target audiences by rapidly devising appropriate and effective methods of communication to distribute essential information, in recognition that traditional media sources may not reach these marginalised communities. Interviewees argued that excluded communities were best reached by those close to them:


*“these are small communities still and, you know, their advocates and their representative groups know where they are”*
(Stakeholder 3—Ireland)

Targeted exposure was reported to have been achieved by way of leaflets or posters placed in strategic position such as shops or doctors’ surgeries, or by using focused social media channels, as discussed above.

A primary criticism of messaging campaigns in some countries concerned the lack of diversity in both language and representation. This had a knock-on effect for services, as illustrated by the next quote from a stakeholder from a UK DA organisation:


*“It wasn’t in other languages. That is why our support workers were ending up with extra work... we had to incur the cost of calling in interpreters to be able to relay that message to our service users. Because, of course, they’re scared, they didn’t know what was happening.”*
(UK Interview 23—Wales)

As highlighted, services working with victims of DA were quick to respond by translating both print and social media materials into various languages. For example, in the UK, a participant in an evidence-gathering webinar reported that various DA organisations collaborated to produce a joint resource available in British Sign Language (BSL) as well as 11 languages for survivors, family members and bystanders. In South Africa, in recognition of the variety of first languages, the Solidarity Fund communicated essential and accurate information messaging in English, Afrikaans, isiZulu, Se Sotho, XiTsonga and TshiVenda [46].

In Australia, an organisation working with CALD communities emphasised the importance of communicating messaging in clear, simple language, including accurate translation of information into various languages, to ensure clear, consistent messaging and avoid confusion for those from CALD backgrounds [52,53]. The importance of providing messaging in a variety of first languages was emphasised by community-based organisations in all participating countries.

Other initiatives, which were developed by services working on the ground with varied minority communities, adopted culturally sensitive approaches to reaching their potential service users. In Ireland, a grassroots Traveller organisation created their own version of the national media campaign ‘Still Here’ which was culturally relevant and accessible for their community. Additionally, this organisation also created a series of videos aimed at Traveller men describing different types of abuse, which conveyed the message that control and abuse are never acceptable; one video had a reach of 5200 [54]. Videos were created and shared over WhatsApp and social media platforms such as Facebook, and featured prominent members of the Traveller community [54]

Meanwhile in South Africa, in spite of the predicted increased demand for shelter spaces during COVID-19, it was reported that when provincial shelter bed capacities were assessed they were found not to have reached anticipated levels [55]. Out of concern, the National Shelter Movement (NSM) publicised their services through a series of radio slots [55]. Additionally, safety plans with provincial NSM representatives’ contact details were circulated on various online platforms [56]. These measures resulted in increased contact with provincial NSM representatives, although it was also reported that counselling rather than shelter was the primary reason for contact at this time [55]. Findings such as these indicate the ability of community-based organisations to reach their target audiences when required. In recognition of the expertise that grassroots and community organisations possess, the Solidarity Fund in South Africa specifically aimed to provide funding to support these community and grassroots organisations during the pandemic in order that they could continue their work [46,57]. In so doing, the Solidarity Fund acknowledged that while some of these organisations may not have had access to traditional funding mechanisms, they were providing critical DA related services on the ground locally [57]. Furthermore, provinces whose DA services were historically underfunded were given special consideration during this process [57].

Lastly, the data collected revealed that, with the exception of Australia, there was a lack of campaigns and messaging aimed specifically at perpetrators of DA during this time. There are earlier examples available of DA campaigns that have targeted perpetrators [58] and targeted campaigns for perpetrators could draw on these examples to signpost where to get support or access behaviour change resources, in addition to highlighting the unacceptable nature of abuse and the human rights violation that DA presents.


*“I think that, as usual, there was not enough or there was kind of under-acknowledgement of the specific ways in which that would change perpetration and affect perpetration, and what was available for perpetrators, although we did see an increase in people [perpetrators] contacting the phone lines.”*
(UK Interview 2—England)

## 4. Discussion

### 4.1. Summary of Outcomes

This exploration of public health messaging in relation to DA during COVID-19 revealed several key areas: Both top-down and bottom-up initiatives were used to communicate messaging. The importance of language and contact with marginalized groups was highlighted. The potential of newer mediums, such as social media and mobile phones, was demonstrated in reaching marginalized groups as was the need for robust evaluation of messaging endeavors. Messaging directed at perpetrators was found to be lacking and this should be considered for future messaging.

### 4.2. Discussion of Findings

Robust top-down responses from government utilising highly visible campaigns across a variety of media during the pandemic were successful in opening up public discourse on DA. [10,14]. It has been argued that, in the context of a pandemic, governments have a duty to adopt “aggressive nationwide campaign[s] to promote awareness about domestic violence through news channels, radio and social media platforms” in order to safeguard human rights, health and victims’ safety [10]. Political leaders possess the ability to influence public opinion, which in turn has the potential to shift attitudes, beliefs and behaviours relating to public health matters [11,12]. Indeed, participants in this study perceived the increased government and media spotlight on DA during the pandemic as overwhelmingly positive. An important impact of this was the perceived heightened awareness of DA created during this time, which was understood to have increased public empathy towards victims. Furthermore, when political acknowledgement of DA was absent from public addresses, it was identified as missed opportunity by participating stakeholders. The data from all countries suggests that participating stakeholders across all nations perceived a clear top-down stance on DA to have been beneficial during the pandemic.

The transparent and regulated approach taken towards government funded advertising campaigns utilised in Australia not only guaranteed that metrics on effectiveness were available; it also ensured that campaign materials were provided in a wide variety of languages, in a culturally sensitive and trauma-informed manner. Meanwhile, in other countries, despite stakeholders’ positive perceptions about national media campaigns, we found little formal evaluation addressing reach or impact. Where evaluation had been undertaken, tangible metrics such as uptake in service use or increased help-seeking behaviour as a direct result of campaigns were unavailable. Due to the high cost associated with large scale media campaigns, and with questions raised about how effective they are in changing behaviours [33], it may be useful to include robust evaluation parameters from the outset which do not solely rely on recall of advertisements as the primary impact metric. Many of the metrics used to measure the effectiveness of social media campaigns such as ‘likes’ and ‘views’ [17] do not provide information on impact.

This study also highlighted the key role that social media and mobile phone messaging played throughout the pandemic in various awareness raising activities. Social media was used as part of large national campaigns to complement traditional media activities and involved generic sharing posts highlighting hashtags, campaign stills or slogans. However, this study has highlighted the targeted use made by community and grassroots organisations of these newer mediums to specifically reach members of their communities who may not be exposed to, or understand, traditional media. WhatsApp and other social media platforms were harnessed by community-based organisations to ‘tailor and target’ messaging to direct reach and enhance impact for specific groups [16]. This indicates the value of increased involvement of community and grassroots organisations to potentially enhance the reach and impact of any future government public awareness campaigns. As with other campaigns, targeted social media activity could benefit from empirical investigation into its effectiveness for dissemination of messaging for DA.

Several studies have highlighted that ethnicity and culture should be considered as important factors when shaping campaigns [13,23,29,31]. Moreover, there is evidence of disproportionate risk levels for marginalised or minority groups (e.g., ethnic minorities, disabled and LGBTQ+ community) arising from the pandemic which exacerbated existing ‘health, socio-political and economic’ inequalities [23,24,25,27,28]. Owing to this disproportionate risk, these minority groups are those who may be in particular need of receiving DA messaging. Recognising and responding to the diversity of audiences is a challenge for generic national campaigns that endeavour to deliver a universal message to what are in fact multiple groups [34]. Special attention to design and format is required to ensure that those who need to receive these campaigns messages are in fact reached and that messages are heard. Lastly, our study identified a lack of national public health campaigns or messaging aimed specifically at supporting DA perpetrators to end their behaviour and seek support, with Australia being the only country in this study do so nationally during the pandemic.

## 5. Conclusions

This paper has examined the approaches used to deliver national messaging in a pandemic across four countries and has identified key strengths and learning to inform future DA public health messaging in times of crisis.

National public health ‘Stay at home to stay safe’ directives used across all four countries in our study were problematic in the context of known rates of DA [59] and with recognition that for victims of DA, home is a very dangerous place [7]. All governments subsequently provided clarity with universal messaging being caveated with a recognition that this only applied if it was safe to remain at home. Governments also used a range of social media formats to provide consistent messaging for DA victims including signposting to support services.

However, key government messages may have failed to reach all those who would have benefited from hearing them since they were not always translated into a sufficient range of languages. We also found that national public health messaging for DA victims were often not translated into appropriate languages or failed to reach diverse communities. This is where national and local DA organisations, as well as partner agencies, stepped in to support the prominence and accessibility of DA messaging for victims. This bottom-up approach, utilising social media as a pivotal tool, enabled diverse audiences to be reached though targeted messages for victims from vulnerable or isolated groups or where language requirements or cultural considerations required addressing. In comparison, we identified very few campaigns aimed at providing DA perpetrators with support to end their behaviour.

Lastly, Australia’s transparent approach to evaluation and auditing of government advertising campaigns could provide a foundational framework for other countries to consider, as we found a lack of robust evaluations of national DA public health campaigns over this period. While participants generally viewed these campaigns as positive, evidence on their reach and impact was missing, limiting our understanding for future crises. Moreover, direction and guidance on public health campaigning can ensure that the learning from individual campaigns is sustained and utilised over time. This study noted the speed with which media coverage and the subsequent public interest can shift to new concerns and causes. For the DA sector, the challenge of maintaining public awareness on DA remains.

However, lessons learnt from the pandemic means we are now acutely aware of how literally people can interpret universal public health messaging, which undoubtably saved lives, but nevertheless had paradoxical and unforeseen consequences for those where home was not safe and provided credence for perpetrators to continue and extend their control over adult and child victims. Future public health responses require a more nuanced understanding of possible unanticipated impacts so these can be recognised and planned for, enabling public health messages to remain clear and accessible for populations while ensuring they do not exacerbate other forms of harm or trauma.

In closing, as this study focused on four high-middle income countries with established DA services, a direction for future research could be to explore countries with disparate economic profiles, or where DA services are not as developed. Indeed, the authors invite researchers from other countries to replicate our approach.

## Data Availability

Data supporting the reported results can be accessed through the ESRC’s repository with the authors’ permission.

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
