# Peer review of "Public Health Directives in a Pandemic: Paradoxical Messages for Domestic Abuse Victims in Four Countries"

_ijerph, 2022, doi:10.3390/ijerph192114148_

Round 1

Reviewer 1 Report

This paper provides a study that is historical timely and has implications for global attention to the impact of the Covid-19 pandemic on the specific issue of domestic violence. It is important to disseminate this kind of overview study of public health issues as quickly as possible. IJERPH is an appropriate place for the work to be disseminated. I have only 3 comments that may strengthening the impact of the paper.

(1) The title and global implications.
This is not really a "global" study. It is a study of 4 wealthy developed countries. I suggest it would be better titled - "Public health directives in a pandemic: Paradoxical messages for domestic abuse victims in four countries". I suggest that the authors make this point in their conclusion, speculate a little on the specific conditions of the four countries, and invite researchers in other countries to replicate their approach. 

(2) The word "This paper" appears very often. Could the authors rephrase some of those sentences. I would prefer "we" but the method of rephrasing is authors' choice. As it is -- the repetitiveness is a little awkward.

(3) Similarly, the "paradoxical message" motif is under-explored in the text and if retained in the title, should be addressed directly.

(4) I think the messages of this study could be strengthened with a summary list of the outcomes at the beginning of the Discussion. The messages could well be emphasised as a set: e.g., top-down and bottom initiatives; the importance of language and contact of marginalised groups; the need for robust evaluation; and especially in this area -- messages directed at perpetrators (I have not made a definitive list here).

Overall, congratulations to the authors in getting this message out and covering vital issues. I recommend timely publication, with a very few pieces of editing.

Author Response

Dear Reviewer 1, 

Please see our response to your review in the attached document. 

Regards, the authors 

Reviewer 2 Report

An interesting article - useful for future similar public health campaigns as you have identified. the two main points is that I didn't find the explanation of methods clear and there are some claims about public health strategies/approaches that are disputed and could be identified as such.   There are some minor comments below. 

Line

32 – I think it would be a good idea/useful to explain why you have chosen the language of domestic abuse – there is different language used all over the world and the choice of this language is interesting – and not language that I am familiar with.  Also, intimate violence – the common form is intimate partner violence – is this not what you mean?  Please explain terms.

34 – omit ‘of’ from the line

49.  Who are the ‘vulnerable populations’? Are you talking about people who are vulnerable to covid or to DV?  If you are talking about DV, the use of vulnerable seems unnecessary – please be specific.

140 – 142. This is a very disputed idea – that telling people things results in behaviour change.  It is certainly not an agreed strategy in health promotion/public health teaching.

162 – 183 – methods.  I find this section confusing.  I don’t understand what you did.

303/304 police were available to provide…

314 – do oyu need ‘also’ in this sentence? Also with who else?

319 – the sentence ‘only made available in some centres’, is a bit confusing until the second read

457 – messaging in English

502/503 – “Robust top-down responses from government during the pandemic succeeded by opening up public discourse on DA and utilising highly visible campaigns across a variety of media”.  There seems to be something missing from the middle of this sentence – did robust top down responses open up highly visible campaigns?

504 – in the context  

Author Response

Dear Reviewer 2, 

Please see our response to your review in the attached document. 

Regards, the authors 
